# Screening BRCA1 and BRCA2 Mutation Carriers for Breast Cancer

**DOI:** 10.3390/cancers10120477

**Published:** 2018-11-30

**Authors:** Ellen Warner

**Affiliations:** Sunnybrook Odette Cancer Centre, University of Toronto, Toronto, ON M4N 3M5, Canada; ellen.warner@sunnybrook.ca; Tel.: +1-416-480-4617

**Keywords:** breast cancer, screening, mammography, magnetic resonance imaging (MRI), sensitivity, specificity, survival, *BRCA1*, *BRCA2*

## Abstract

Women with BRCA mutations, who choose to decline or defer risk-reducing mastectomy, require a highly sensitive breast screening regimen they can begin by age 25 or 30. Meta-analysis of multiple observational studies, in which both mammography and magnetic resonance imaging (MRI) were performed annually, demonstrated a combined sensitivity of 94% for MRI plus mammography compared to 39% for mammography alone. There was negligible benefit from adding screening ultrasound or clinical breast examination to the other two modalities. The great majority of cancers detected were non-invasive or stage I. While the addition of MRI to mammography lowered the specificity from 95% to 77%, the specificity improved significantly after the first round of screening. The median follow-up of women with screen-detected breast cancer in the above observational studies now exceeds 10 years, and the long-term breast cancer-free survival in most of these studies is 90% to 95%. However, ongoing follow-up of these study patients, as well of women screened and treated more recently, is necessary. Advances in imaging technology will make highly sensitive screening accessible to a greater number of high-risk women.

## 1. Rationale for Breast Screening

For the woman with an inherited *BRCA1* or *BRCA2* mutation who wishes to do everything possible to avoid developing breast cancer, risk-reducing mastectomy at age 25 is undoubtedly the optimal strategy [1]; however, for many mutation carriers, other factors may be of equal or greater importance. Some women wish to postpone preventive surgery until they have found an intimate partner, finished childbearing and breast-feeding, or reached an age when their cancer risk becomes unacceptably high. Many others decline surgery in the hope they will be fortunate enough to never develop breast cancer but would opt immediately for bilateral mastectomy if cancer were diagnosed, regardless of its stage. Others choose breast conservation after a breast cancer diagnosis and, perhaps surprisingly, a sizable minority are so averse to mastectomy that they opt for repeat lumpectomy (generally with re-irradiation) even after an ipsilateral cancer recurrence. For all these women who wish to defer or altogether avoid preventive surgery, a breast screening regimen that can reliably detect breast cancer at a stage when the probability of cure is very high is essential.

## 2. Screening Mammography

For the last 50 years, mammography has been the gold standard for breast cancer screening for women in the general population and is the only breast imaging modality that has been proven to reduce breast cancer mortality in randomized controlled trials [2]. Accordingly, with the identification in the mid-1990s of women who carried *BRCA1* and *BRCA2* mutations, annual mammography (along with semi-annual clinical breast examination and monthly breast self-examination) became the recommended screening regimen for these women [3]. However, prospective studies showed that with this approach a large proportion of cancers were detected at a suboptimal stage. The interval cancer rate ranged from 35% to 50%; few cases of ductal carcinoma in situ (DCIS) (the ideal stage for detecting cancers as cure rates approach 100%) were found; 40% to 78% of the invasive cancers were greater than 1 cm in size, and 20% to 56% of the invasive cancers had lymph node involvement [4,5,6,7].

There are likely several factors accounting for the low sensitivity of mammography in these studies. Young women generally have greater radiologic breast density (x-ray-attenuating fibroglandular tissue) than older women, which obscures the detection of malignancy [8]. However, even among women of a given age group, *BRCA1*-related cancers in particular have been shown to be less mammographically detectable [9,10]. Histologically *BRCA1*-related cancers are typically cellular and fleshy with round pushing margins (atypical medullary histology), resulting in a more benign mammographic appearance, rather than scirrhous with irregular infiltrating margins like many sporadic cancers [11]. In addition, *BRCA1*-related tumors are less likely to be associated with significant amounts of DCIS [12], the latter often developing microcalcifications that lead to detection by mammography. The lower sensitivity of mammography for *BRCA2*-related cancers is somewhat more difficult to explain, as histologically these cancers are similar to sporadic cancers. A reasonable explanation is that, on average, *BRCA2*-related cancers (like *BRCA1*-related cancers) have a faster doubling time than sporadic cancers [13], which increases their likelihood of presenting as palpable interval cancers rather than being detected on the next round of screening.

## 3. Magnetic Resonance Imaging (MRI)

Unlike mammography, which relies on anatomic density, distortions, and calcifications in order to detect malignancy, contrast-enhanced magnetic resonance imaging (MRI) provides a functional assessment of breast tissue. Its ability to detect tumor neovascularity and peritumoral inflammation renders its sensitivity relatively independent of breast density and higher than that of any other breast imaging modality [14]. Approximately 50% of invasive cancers demonstrate a classic MRI pattern of early contrast enhancement and early washout, because the contrast agent accumulates faster and washes out faster from the more vascular tumor than from the normal or benign tissues. Certain morphologic features are also typical of malignancy including spiculated or irregular lesion margins and increased enhancement of the lesion’s periphery. Absence of enhancement correlates well with absence of invasive breast cancer, with a negative predictive value of over 95% [15].

Not surprisingly, given the observed low sensitivity of mammography for *BRCA*-related cancers and reports of the high sensitivity of breast MRI as a diagnostic tool, multiple non-randomized observational screening studies were started in the mid to late 1990s in which concurrent annual MRI was added to mammography for screening *BRCA* mutation carriers, with the performance of each imaging modality evaluated independently. A meta-analysis of the results of these studies [16] found that the sensitivity of mammography ranged from 25% to 59% with a pooled sensitivity of 39% (95% CI 37–41), while MRI had a sensitivity of 68% to 100% with a pooled sensitivity of 77% (95% CI 70–84) using a cut-off value of BI-RADS 3 or higher to define a positive study. The two modalities were complementary, as the sensitivity of the combination was 94% (95% CI 90–97). Most importantly, the majority of cancers detected in these studies were either non-invasive or very early invasive cancers with a node-positive rate of 12% to 26%. In view of these highly consistent results, annual MRI has now been incorporated into all recent breast screening guidelines for *BRCA* mutation carriers [17,18,19,20] (see Table 1).

There are several barriers to the widespread availability of screening MRI. Its very high cost is a major limitation for middle-income and developing countries. But even in more affluent countries, screening MRI is not universally available. A dedicated breast coil, capacity for MRI-directed biopsies, and radiologists experienced in reading breast MRI must all be available for a center to be able to offer reliable screening MRI. It has been suggested that high-risk screening centers should offer at least 150 screening breast MRI examinations per year and perform at least 10 MRI-guided breast biopsies [21]. Centers that lack the capability of performing MRI-guided biopsies should not offer breast screening with MRI (assuming MRI-guided biopsy is available at another centre within a reasonable commuting distance), as the major benefit of MRI is its ability to detect non-calcified DCIS and tiny invasive lesions not visible with other modalities.

Even when availability is not an issue, MRI has several drawbacks, the biggest of which is its relatively low specificity that precludes its use as a screening tool for the general population. In the above-mentioned meta-analysis, the false positive (recall) and biopsy rates averaged over all rounds of screening were 13.7% (95% CI 8.3–19.1%) and 3.9% (95% CI 2.6–5.2), respectively for MRI compared to 5.3% (95% CI 3.5–7.0%) and 1.5% (95% CI 0.8–2.2) for mammography [16]. Notably, recall and biopsy rates were generally substantially higher in the first year of screening in the absence of a previous MRI study for comparison, dropping by approximately one-third on subsequent rounds of screening [22]. These false positive rates also tended to be higher in North American centers than in Europe [16]. It appears that relatively high false positive rates are generally considered acceptable when screening very high-risk populations; the false positive rate for the MRI screening studies compares favourably with the 12.9% to 25.9% false positive rate reported over the first three screening rounds of the National Lung Screening Trial [23].

Additional drawbacks of MRI include: the need to perform the test during the second week of the menstrual cycle to optimize sensitivity and specificity; the need for an intravenous line to inject the gadolinium-based contrast agent; claustrophobia in up to 10% of women (though usually amenable to a mild sedative); and contraindications such as indwelling metal devices and renal failure.

While, until recently, possible morbidity from the contrast agent was thought to be limited to patients with renal dysfunction who are at risk for the relatively rare syndrome of Nephrogenic Systemic Fibrosis [24], since 2014, numerous clinical studies have shown long-term retention of gadolinium in the brains of subjects with normal renal function after repeated contrast-enhanced scans, especially in the dentate nuclei of the cerebellum and in the globus pallidus of the basal ganglia. This retention is more pronounced with the five currently used linear gadolinium-based contrast agents than with the three macrocyclic ones [25]. Retention of gadolinium in bone, skin, and other sites has also being recently reported; however, to date, no neurological or other clinical consequences of gadolinium retention have been observed in patients with normal renal function. [26].

Given the expertise necessary to perform and interpret screening breast MRI, one might wonder whether the excellent results reported by the observational studies conducted, for the most part, at large academic research centers would be achievable in a community setting. The Ontario High-Risk Breast Screening Program is a population-based screening program in Canada’s most populous province for high-risk women (known mutation carriers or calculated lifetime breast cancer risk of 25% or higher) that was established in 2011 and is currently available at 28 centers, the majority of which are non-academic. Eligible patients aged 30 to 69 undergo annual mammography and MRI screening [27]. Five-year results for this program have recently been analyzed, and a manuscript is in preparation.

## 4. Is MRI Sufficient for Breast Screening?

Considering the very high sensitivity of MRI and the very low sensitivity of mammography for *BRCA* mutation carriers in the screening setting, one wonders whether there is any point adding screening mammography to MRI. Furthermore, at least in theory, annual mammography started at a very young age might actually be harmful. All cells of germline *BRCA* mutation carriers have subclinical defects in DNA repair which could render those cells more susceptible to the mutagenic effects of ionizing radiation [28]. To date, studies to determine whether there is a correlation between diagnostic radiation and breast cancer risk in *BRCA* mutation carriers have yielded conflicting results. In a very large European study of female *BRCA1* and *BRCA2* mutation carriers that attempted to correct some of the methodological limitations of prior studies, exposure to diagnostic radiation including mammography before the age of 30 was associated with an increased risk of breast cancer (HR 1.9, 95% CI 1.2 to 3.0), at dose levels significantly lower than those at which increased risk has been found in non-carriers. No increased risk was found for exposure at ages 30 to 39 [29]. On the basis of this study, NCCN guidelines no longer recommend screening mammography before age 30 [18]. In contrast, in a study of 2346 mutation carriers who recorded their history of mammographic screening at study entry and were followed prospectively for the development of breast cancer, after correcting for other risk factors, no correlation was found between prior mammography exposure before age 30 and breast cancer incidence [30].

Even if mammography is not harmful after age 30, is it beneficial? In a report from one centre in the Netherlands, of 94 breast cancers diagnosed in *BRCA1* mutation carriers subsequent to the introduction of digital mammography, 88 cases were diagnosed by annual MRI but only two cases were detected by mammography alone, both of these DCIS in patients over age 50. The authors recommended that screening mammography need not be added to MRI in this population before age 40 [31]. A very recent report from a high-risk screening program at another centre in the Netherlands performing annual MRI plus mammography confirmed these findings. Among 61 cancers diagnosed in *BRCA* mutation carriers, only three (one invasive and two DCIS) were detected by mammography alone, and all three were in women aged 50 or older [32].

In the Italian muticentre HIBCRIT-1 screening study, mammography did not significantly increase cancer detection rates compared to MRI alone for either *BRCA1* or *BRCA2* mutation carriers. The investigators concluded that mammography should not be used to screen *BRCA* mutation carriers outside a clinical trial. An ongoing trial in Italy is currently randomizing high-risk women to be screened with either MRI alone or with MRI plus ultrasound up to age 35 and MRI plus ultrasound and mammography after age 35 [33].

In the meantime, the European Society of Medical Oncology and the United States National Comprehensive Cancer Network continue to recommend screening mammography starting at age 30 [18,19]. Radiologists advocating this approach argue that mammography remains the only screening modality proven in randomized trials to reduce breast cancer mortality. However, there has never been a randomized trial of MRI screening, and such a trial would no longer be either ethical or feasible. Radiologists further point out that mammography sometimes detects low-grade lesions not visible with MRI and that the performance of mammography concurrently with MRI helps with MRI image interpretation. Despite this last point and the fact that in all the observational studies MRI and mammography were performed concurrently, it is customary in many centers to alternate annual mammography and MRI at six-month intervals in order to maximize the opportunity to diagnose very rapidly growing cancers that might otherwise present as interval cancers. Although this practice might be reassuring to patients and referring physicians, there is so far no robust data supporting its superiority over concurrent imaging. A study evaluating this issue is underway [19].

Because screening ultrasound often detects invasive cancers missed by mammography, particularly in women with dense breasts, several of the observational screening studies of annual MRI plus mammography also included screening ultrasound [22,34,35,36,37] and/or clinical breast examination [22,34,37,38,39]. Neither ultrasound nor clinical breast examination significantly increased the cancer detection rate, but each additional modality did increase the number of false positives. Nonetheless, clinical breast examination performed every 6 to 12 months and ‘breast awareness’ continue to be recommended by many experts for screening *BRCA* mutation carriers as an adjunct to imaging [18,19]. There is general agreement that annual screening ultrasound (performed together with annual mammography) should be reserved for women who are unable to access or tolerate MRI (even with sedation) or for whom MRI is contraindicated [17,19,20,40]. A summary of the various expert screening recommendations can be found in Table 1.

Although ultrasound has no role as a screening modality if MRI can be performed, it has a critical role in the investigation of mass lesions detected by screening MRI. Ultrasound can detect approximately two-thirds of lesions found by MRI and may be able to give a definitive diagnosis of a benign lesion such as a cyst or fibroadenoma without the need for biopsy. Lesions are much more likely to be sonographically detectable if they are masses, particularly large ones, rather than non-mass enhancement. Biopsies can also most easily be performed under ultrasound guidance if the lesion is visible sonographically. Although the risk of malignancy has been reported to be lower for lesions that are sonographically occult than for those with a non-benign sonographic correlate, suspicious lesions seen only with MRI must be biopsied under MRI guidance [41].

## 5. Age to Start and Stop Screening and Optimal Screening Interval

On the basis of the known age-related incidence of cancer in BRCA mutation carriers, it is recommended that MRI screening begin at age 25 [18,19] or 30 [17,20]. Some experts have advocated ‘breast awareness’ as early as age 18 to 20 [18]. While this may be reassuring to anxious physicians and parents, there is no available evidence that it is effective.

At the time that screening MRI was first being studied in *BRCA* mutation carriers, it was hypothesized that MRI would likely be necessary only up to age 50, the age from which mammography has high sensitivity in the general population due to the progressive decline in average breast density with age. Surprisingly, in those studies that enrolled women beyond age 50, the benefit of adding MRI to mammography was at least as great for participants aged 50 and over as it was for younger women [42]. This suggests that the low sensitivity of mammography in *BRCA* mutation carriers is not simply a function of the generally greater breast density of younger women. This is supported by the finding in the Toronto study that, although screening mammography was more sensitive in BRCA mutation carriers with fatty breasts than in those with greater breast density, mammography failed to detect half of the cancers in women whose mammographic density was described as ‘fatty’ or ‘scattered fibroglandular density’ [43]. The rationale for the NICE guideline [17] for not continuing MRI after age 50 for women who do not have dense breasts is, therefore, unclear. Since none of the screening studies enrolled a significant number of participants above age 60, it is impossible to know when MRI can be safely discontinued beyond that age. The rationale for discontinuing MRI screening after age 70 in the High-Risk Ontario Breast Screening Program [27] was that the sensitivity of mammography improves with age [44] while the growth rate of *BRCA*-related cancers slows with age [13], and that breast screening has not demonstrated a mortality benefit in women over age 69 in the general population. The NCCN, however, recommends that MRI be continued until age 75 [18], while ESMO does not specify a cut-off age [19].

Since in many studies the interval cancer rate was highest in *BRCA1* mutation carriers under age 50, [22,45,46,47], it has been suggested that, because of the higher growth rate of cancers in this population [13], perhaps MRI should be done every 6 months in this sub-group [46]. Given the obvious significant cost and resource implications of such a policy, unless the superiority of more frequent screening is proven, annual screening will remain the standard. On the other hand, it might be argued using the same reasoning that, with the known slowing in the growth rate of *BRCA*-related breast cancers with age (as with sporadic cancers) [13], MRI screening could safely be done less frequently—perhaps every 2 years—in women over age 60 who continue to undergo annual mammography. This argument appears even stronger for *BRCA2* mutation carriers whose cancers often remain in situ for several years. While such an approach requires formal testing, in the interim it would be a reasonable policy to adopt in a setting with limited resources. One fact that has been established, however, is that once MRI has been discontinued, screening mammography must be continued on an annual basis. In a case–control study in the Netherlands, *BRCA* mutation carriers aged 60 and over screened every two years with mammography, as per their national guidelines, had twice the interval cancer rate and were 2.5 times as likely to have unfavorable breast cancer histology as mutation carriers who underwent mammography yearly [48].

Although one of the advantages of risk-reducing mastectomy is the elimination of the need for further breast screening, some women feel uncomfortable with this recommendation, particularly if they have had implant reconstruction. This likely relates to their awareness that the surgery often leaves behind microscopic amounts of breast tissue on the chest wall or in the axilla and their worry that, should a cancer develop, it might be masked by the implant. Expert guidelines recommend against screening this population for breast cancer [17,20]. These women need to be reassured that, in the rare event that breast cancer would develop after this surgery, it would be easy to detect under the skin. Furthermore, they should be told that, because implants are inserted behind the pectoralis major muscles, there is no breast tissue behind the implants.

## 6. Survival of Screened *BRCA* Mutation Carriers Who Develop Breast Cancer

Because MRI screening trials for *BRCA* mutation carriers only began in the late 1990s and because breast cancers often metastasize more than a decade after diagnosis, none of these trials have sufficient long-term survival data for one to accurately determine the cure rate of breast cancers in a population screened with annual MRI. In the Dutch MRISC study, five (10%) of the 51 *BRCA* mutation carriers developed distant recurrences at a median follow-up of nine years, with no difference between *BRCA1* and *BRCA2* mutation carriers [49]. In the combined UK MARIBS and NICE studies in which 45 breast cancers were detected, only two deaths (both in BRCA1 mutation carriers) were observed at a median follow-up of 12 years [50]. The percentage of women who had developed metastatic disease was not stated. In contrast, a study from Norway reported 10 deaths in 68 BRCA1 mutation carriers at a median follow-up of 4.2 years for an estimated five-year breast cancer specific survival of only 75% and 10-year survival of 69% [51]. These relatively poor results can probably be explained by the lower sensitivity of MRI in that study compared to other studies and the fact that the centres lacked the capability of performing MRI-guided biopsies. The longest reported follow-up to date from the observational screening studies is from the Toronto study. At a median of 14 years, in 40 previously unaffected women who developed breast cancer during the screening period, only two breast-cancer related distant recurrences (and deaths) were observed, at six and seven years from diagnosis. Interestingly, there were four additional deaths from other causes, three of them cancer-related [52].

For health care policy-makers, the critical question is whether the high cost of screening MRI translates into a significant survival benefit compared to screening mammography alone, or whether it simply provides lead time. Since there will never be a randomized controlled trial of screening with or without MRI, the best one can do to answer this question is to compare the long-term outcome of a cohort of mutation carriers screened with mammography alone to that of a matched cohort screened over the same time period with both modalities. The preliminary data are encouraging. In a prospective trial, previously unaffected *BRCA* mutation carriers screened with breast MRI in the Toronto study had tumors detected at a significantly earlier stage than matched mutation carriers screened with mammography alone [53]; however, the follow-up is still too short to determine whether this will ultimately translate into a significant distant recurrence and survival difference. In a retrospective UK study of *BRCA* mutation carriers, 10-year survival was 95.3% for those women screened with MRI plus mammography compared to 87.7% for a matched cohort only screened with mammography (HR 0.21, *p* = 0.03) [50].

For patients, the long-term survival data are even more critical. Although many patients choose risk-reducing mastectomy over breast screening in order to avoid the anxiety of frequent tests, false positives, and the high likelihood of a cancer diagnosis followed by the morbidity of treatment, for many other patients screening would be the preferred option if they could be assured that cancer would, with very high likelihood, be detected at a curable stage. A 100% cure rate is neither realistic nor necessary as, even with risk-reducing mastectomy, there is still some risk of developing (potentially fatal) breast cancer as well as a not insignificant risk for the development of other cancers. Since the ideal randomized trial of screening versus risk-reducing surgery will for obvious reasons never be done, indirect comparisons are necessary. A Monte Carlo computer simulation model estimating survival among *BRCA1* and *BRCA2* mutation carriers undergoing different risk-reducing strategies found that the combination of risk-reducing mastectomy and risk-reducing salpingo-oophorectomy at age 30 provided the greatest gains in life expectancy, but that substituting intensive breast screening for mastectomy would only reduce life expectancy by a maximum of 1.5 years for BRCA1 mutation carriers and 0.7 years for BRCA2 mutation carriers [1]. A recent German study also found only a loss of 0.5 life-years with salpingo-oophorectomy alone compared to risk-reducing mastectomy plus salpingo-oophorectomy at age 30 [54]. 

## 7. Novel Imaging Modalities

The high direct and indirect costs of MRI limit clinical access to screening MRI. One of the factors accounting for this high cost is the long time it takes to acquire MR images (20 to 40 min) and to read the hundreds of images that are generated. Recently, several groups have reported results of an abbreviated MRI protocol (AP) limited to the early post-contrast period, followed by a more rapid overview of the imaging volume with the acquisition of far fewer images than a standard protocol. Five of six studies have shown that an AP can shorten both acquisition (average of nine minutes, range 3–15 min) and reading time, while retaining diagnostic accuracy [55].

One of the limitations of an AP is the potential lack of specificity due to the absence of kinetic assessment which requires the multiple sequential sets of post-contrast images omitted by the AP. Unlike the AP, which is merely a shortened version of the conventional MRI protocol, a newer technique called ‘Accelerated MRI’ or ‘ultrafast MRI’ acquires each image more quickly than the conventional method, thereby reducing the acquisition time while retaining both high spatial and high temporal resolution [56]. However, both abbreviated and accelerated protocols have yet to be validated for screening a population of BRCA mutation carriers.

Contrast-enhanced digital mammography (CEDM) is a promising new imaging modality that might be an excellent option in centres where MRI is unavailable or for women with a contraindication to MRI. Although digital mammography is superior to film-screen mammography, particularly in women under 50 and those with dense breasts, it still suffers from the problem of projecting a three-dimensional object onto a two-dimensional surface, resulting in the masking of tumors or false positives from the overlapping of densities. Breast tomosynthesis, which creates a three-dimensional breast image from a stack of multiple 2D slices, is significantly more sensitive and specific than digital mammography, as it separates many of the overlapping densities [55]. CEDM, on the other hand, like MRI, provides a functional evaluation of tissue neovascularity achieved by the injection of an iodinated contrast material. In a recent study of 318 high-risk women, three cancers that were occult on conventional digital mammography were detected by MRI, and two invasive cancers were also detected by CEDM [57]. Contrast-enhanced breast tomosythesis can also be performed and might have even greater sensitivity and specificity [58]

## 8. Conclusions

MRI-based breast screening is currently a very reasonable option for female *BRCA* mutation carriers who wish to delay or avoid risk-reducing breast surgery. Moreover, MRI technology will undoubtedly improve over the coming years, becoming more effective and more affordable. In parallel, with treatment advances such as PARP inhibitors that target *BRCA*-related cancers, the prognosis of screen-detected breast cancers will undoubtedly become even better over the coming years. Understandably, many women who carry *BRCA* mutations find their high risk of breast cancer unacceptable and opt for risk-reducing surgery over screening, even if they can be assured that any cancer that developed would likely be detected at a curable stage. Hopefully, in the not too distant future, we will be able to offer *BRCA* mutation carriers effective, safe, and well-tolerated, non-surgical risk reduction combined with a safe, highly sensitive, and specific screening regimen.

## Figures and Tables

**Table 1 cancers-10-00477-t001:** Expert recommendations for screening *BRCA* mutation carriers for breast cancer.

Organization	Annual MRI	Annual Mammography	Screening Ultrasound	Other
NCCN [18] 2018 (U.S.)	Aged 25–75	(**with consideration of tomosynthesis**) Aged 30–75 Aged 25-75 if MRI not possible	Not recommended	Breast awareness aged 18+ Semi-annual CBE aged 25+
NICE [17] 2017 (U.K.)	Aged 30–49 Aged 50–69 only if mammo-graphically dense breasts	Aged 40–69	Aged 30–49 if MRI not possible	Breast awareness
ESMO [19] 2016 (Europe)	Aged 25+	Aged 30+	Aged 25+ if MRI not possible	Breast awareness Semi-annual CBE aged 25+
CCO [20] 2018 (Canada)	Aged 30–69	Aged 30+	Aged 30–69 if MRI not possible	Breast Awareness

CBE: clinical breast examination.

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
