# Peer review of "Screening BRCA1 and BRCA2 Mutation Carriers for Breast Cancer"

_cancers, 2018, doi:10.3390/cancers10120477_

Reviewer 1 Report

This is a generally well written overview.

There are defined guidelines on screening women with BRCA mutations (NCCN, ESMO and NICE), so summarising their key features in a table would be easier for readers to see them and allow some comparison between them (particularly for mammography, MRI and ultrasound).  [on line 144, reference 15 should be 16].

The Abstract contains results from the Ontario screening programme (cancer-free survival 90-95%), but these do not appear anywhere in the article. Furthermore, if specific attention is to be made to this program, it would be appropriate to include a simple table showing the survival results of this and other similar large-scale programs, particularly given that the full Ontario results have not been published yet, and that this article is meant to be a review.

The higher false-positive rate using MRI is described as a major limitation, but actually might be acceptable when dealing with particularly high risk individuals (and 13.7% is not too dissimilar to that for low dose CT screening for lung cancer in the large US trial, NSLT).

Author Response

Response to the academic reviewer is below.  Response to Reviewer 1 is attached.

1)    pls check for spelling and grammar and errors.

 I have corrected all that I found

2) suggest removing the material on the risk of early mammography in brca1 carriers or add the reference to giannakeas et al. about mammography in carriers.  breast cancer research and treatment august 2014  

 I have added the Giannakeas reference and briefly mentioned the key findings. 

Reviewer 2 Report

Line 29-30 please provide reference for this statement.

Line 46, also please provide reference.

Please reformulate the concept involving BRCA2 mutated cancers and mammography in lines 63-65.

How is the pooled sensitivity calculated in line 82-83?

Could you please briefly discuss screening post mastectomy?

Line280: is that statement correct: substituting intensive breast screening for mastectomy would reduce life expectancy..?if not, could you reformulate the sentence?

possible typos:

line 38: with-->wish

line 187; missing "has"

line 295: "protocol a .."

I have not used any program to check for plagiarism so I cannot be 100% sure on the plagiarism checkings

Author Response

Comments and Suggestions for Authors

1) Line 29-30 please provide reference for this statement.

Reference 1 has been added 

2) Line 46, also please provide reference.

Reference 3 has been added

3) Please reformulate the concept involving BRCA2 mutated cancers and mammography in lines 63-65.

This has been rewritten for clarity.

4) How is the pooled sensitivity calculated in line 82-83?

I explained that the cut-off for positive was a BI-RADS score of 3 or higher

5) Could you please briefly discuss screening post mastectomy?

I have added a brief paragraph about this.

6) Line280: is that statement correct: substituting intensive breast screening for mastectomy would reduce life expectancy..?if not, could you reformulate the sentence?

This is correct according to the model used in these publications. 

 7) possible typos:

line 38: with-->wish

line 187; missing "has"

line 295: "protocol a .."

The typos have been corrected.